# Determinants of Household-Level Food Storage Practices and Outcomes on Food Safety and Security in Accra, Ghana

**DOI:** 10.3390/foods11203266

**Published:** 2022-10-19

**Authors:** Ebenezer Afriyie, Franz Gatzweiler, Monika Zurek, Freda E. Asem, John K. Ahiakpa, Bernard Okpattah, Emmanuel K. Aidoo, Yong-Guan Zhu

**Affiliations:** 1Institute of Urban Environment, Chinese Academy of Sciences, Xiamen 361021, China; 2University of Chinese Academy of Sciences, Beijing 100043, China; 3Environmental Change Institute, University of Oxford, South Parks Road, Oxford OX1 3QY, UK; 4University of Ghana, Legon P.O. Box LG 25, Accra, Ghana; 5Research Desk Consulting Limited, Kwabenya P.O. Box WY 2918, Accra, Ghana; 6Asinyo Agri-Commerce Ltd., Cantonments P.O. Box CT 9823, Accra, Ghana

**Keywords:** food storage, food waste, food shopping, food commodity, Ghana, food security

## Abstract

Household-level food storage can help families save money, minimize food waste, and enhance food safety and security. Storing food within households may, however, be affected by domestic routines, like food shopping and cooking. Therefore, it is essential to evaluate how consumers’ attitudes and behaviors influence food storage at the household level. This study aimed to assess the determinants of household-level food storage, ascertain consumers’ behavior and perception towards food storage, and assess the effect of household-level food storage on food safety, wastage, food expenditure and security. Dzorwulu and Jamestown, both located in Accra, Ghana, served as the study’s primary sites. The study employed a survey and structural equation modeling to evaluate key determinants of household-level food storage practices and their impacts. A semi-structured questionnaire was administered to 400 food household heads, sampled using a systematic sampling procedure. The results showed that food shopping drives food storage. There was, however, significant negative association (p < 0.001) between food shopping and time of food storage. Although cooking impedes household-level food storage, there was significant positive association (p < 0.001) between frequency of cooking and storage period of food commodities. The findings also revealed that household-level food storage promotes food safety, reduces food expenditure and waste, and contributes to enhancing food security by 43%. To promote household-level food storage and ensure food safety and security, future study should concentrate on enhancing conventional household-level food storage practices that are efficient, cheaper and easily implementable.

## 1. Introduction

Food storage enables food to be kept or stocked up in a designated and suitable storage space for future consumption [1]. Storing food is critical for food security and has various purposes, including enabling a better-balanced diet throughout the year, since it helps to maintain the nutritional value and quality of foods [2]. Food storage facilitates the distribution of harvested and processed foodstuff to consumers. For instance, meat can be distributed to different locations within 8–14 weeks of transportation, when stored at −1.5 °C in refrigeration systems of transport containers, and still maintain its quality [3]. Storing food reduces food waste at the household level by preserving leftovers, unused or uneaten food for later use. A study by [4] found that 12% of food eaten by respondents were leftovers and 24% of food not fully eaten were stored as leftovers to be eaten later. It is also important to store food for periods of scarcity, catastrophes and pandemics. The COVID-19 pandemic, for instance, caused panic buying and storing of food among households, due to restrictions imposed by various countries around the world on the movement of people and goods [5,6].

Household-level food storage is essential for helping households avoid food losses, cope with growing food demand and improve food security [7]. Food security is defined as when “all people, at all times, have physical and economic access to sufficient, safe and nutritious food that meets their dietary needs and food preferences for an active and healthy life” and it is achieved when all of its four dimensions, including food availability, food access, utilization of food and food stability, are realized [8]. The World Bank, in 2017, reported that 83 million people in 45 countries were starving [9]. According to the state of food security and nutrition in the world report, the number of hungry people in the world rose from 633.4 million in 2018 to 768 million in 2020 [10]. With the continuous increase in population, the possibility of eliminating hunger by the year 2050 is becoming challenging [11]. Most of the world’s human population is currently living in cities and urban population is projected to increase to about 6 billion people by 2050 [12]. Feeding this growing urban population will involve increasing agricultural production by 70%, which may be a challenge, due to declining arable land, spurred on by competing demands from industrialization and urbanization [13]. Nevertheless, promoting food storage, especially at the household level, can help to avoid wasting or losing what can be produced, and make food adequately available to urban households.

Ready access to safe and nutritious food is an important basic human right. However, more than 420,000 people die, and about 600 million people fall sick, yearly after eating contaminated or unsafe foods [14]. Food safety is about handling, preparing and storing food in order not to cause infections or diseases to the consumer when the food is eaten, and also to ensure that food contains adequate nutrients [15]. Consumer behaviors toward food are currently changing in response to factors, including climate change, change in incomes, attention to improving health, and concerns about ensuring environmental sustainability in food production [16,17]. These factors are associated with potential food safety and food security threats, drivers or inhibitors that need to be evaluated to protect the health and wellbeing of consumers. There have been several cases of foodborne illness due to issues of unsafe food practices by consumers. In order to minimize the risk of foodborne illness, consumers must be ready to change their behaviors and attitudes towards unsafe food storage, handling and preparation practices [18]. Ensuring proper food safety practices is beneficial for sustaining life and enabling healthy diets, preventing foodborne diseases, producing and preparing food safely, promoting trade and access to new markets, and minimizing food loss and waste [19]. Storing food at the household level, which is mainly the decision of consumers, is influenced by their knowledge on food storage, and available food storage technologies and behaviors, such as feeding, cooking, food shopping, and food choice motives, including food pricing [20,21]. Some drivers or variables, such as household size, number of children, frequency of shopping trips, lack of planning and not adhering to food safety standards, have been identified by some studies to be related to consumer food storage decisions at the household level [22,23,24].

In Ghana, approximately 5% of the population is regarded as being food insecure and about 6.7% as vulnerable to become food insecure, which means that an unexpected shock would have huge impact on their food consumption patterns [25]. The findings of a study by [26] showed that about 70% of sampled households (n = 668) were low-income and middle-income residents of Accra, Ghana, and were severely food insecure and that households were mostly worried about not having sufficient food and, occasionally, not being able to access enough food to meet their needs. Furthermore, it has been reported that Ghana faces several food safety challenges, all of which have adverse effect on public health and wellbeing, including the following: mycotoxin contamination; microbial contamination; mercury in fish; polycyclic aromatic hydrocarbons (PAH) in smoked meat and fish; pesticide residues in legumes, grains, fruits and vegetables; food adulteration; and inappropriate use of food additives [27]. Consumers in Accra raised concerns with regards to foodborne diseases, including typhoid, cholera, diarrhea, food poisoning, swine flu and bird flu, according to a research by [28]. Diarrheal disease was reported to be the fourth cause (7% of total cases) of morbidity and among the twenty highest causes of outpatient morbidity in 2016 in Ghana [27]. A research by [29] in the Greater Accra Region of Ghana also showed that up to 20,199 cholera cases were recorded, with about 50% of the cases occurring in the Accra metropolitan area. It is, therefore, critical to evaluate various ways of improving food security and food safety, particularly within households, considering the severity of the impact of food insecurity and food safety risks in Ghana. In this paper, we assessed the determinants of household-level food storage practices and outcomes on food safety and security among households in Accra, Ghana. Specifically we performed the following: (a) identified the drivers and inhibitors of food storage practices among households, (b) assessed the behavior and perception of households towards household-level food storage, (c) ascertained the effect of household-level food storage on food safety, (d) assessed the impact of household-level food storage on food wastage within households, (e) ascertained the effect household-level food storage has on household food expenditure, and (f) assessed the effect of household-level food storage on perceived food security.

## 2. Materials and Methods

### 2.1. Study Area

This work forms part of a larger study conducted in Accra, the capital city of Ghana. The city has a population of more than 2 million people and occupies a land area of 225.67 square kilometers [30]. More than 50% of the residents in Accra are migrants and about 38.4% live in communities regarded as slums, where urban poverty may be considered as endemic [31]. Two communities in Accra, which are those of Dzorwulu and Jamestown, were selected for this study. Selection of these two communities was due to the socio-economic statuses of their residents. Dzorwulu has 3309 households and consists of inhabitants who are regarded as mainly of middle-income status [32,33]. Jamestown has 5013 households, and is considered to be a low-income community because of its low socio-economic status compared to the national average. Residents live in poor or congested housing and have low levels of education [34,35].

### 2.2. Data Collection

This study consisted of a survey which was done by administering semi-structured questionnaires to food heads of households over 10 weeks, from November 2020, to January 2021. The survey method has been used in various studies to assess food safety, security, expenditure and food waste of households or consumers [26,36,37,38,39]. The food head of a household is the individual who has the major responsibility of planning, shopping and preparing food for members of the household at home [40]. The survey questionnaire consisted of 47 questions and was designed such that it would need a maximum of 40 min to complete. Overall, 400 questionnaires were administered and all were valid, representing a 100% response rate. The questionnaire sought to find out respondents’ household food storage behavior and practices, and their influence on food safety, security, expenditure and waste. It covered demographic characteristics, knowledge of food storage, cooking and feeding behavior, food purchase behavior, food choice motives, food storage behavior and practices, food handling and packaging, food waste management, food safety and health, food expenditure and food security. Before conducting the survey, the questionnaire was pre-tested on 35 households at Osu, another community in Accra with characteristics similar to the two study communities. Pre-testing the questionnaire was done to avoid ambiguity, ensure participants understood its content and, also, to estimate the amount of time that would be needed to answer all the questions. The survey was carried out in the language preferred by the participants, English was preferred in Dzorwulu, while Ga and Akan (local languages) were preferred in Jamestown. The survey was done via face-to-face interviews by three interviewers who spoke English, Ga and Akan languages, and were trained and supervised to collect the data.

The total number of respondents used for this study was 400; Jamestown had 240 respondents, while Dzorwulu had 160 respondents. A stratified random sampling procedure (proportional allocation) was used to select respondents from the two communities using Equations (1) and (2). Stratified random sampling reduces the potential for human bias and, hence, provides a sample that is representative of the study population, supposing there is limited missing data. The method is, however, not useful when the population cannot be divided into disjoint subgroups exhaustively [41]. Household selection was done using the systematic sampling procedure, which has the merit of spreading the sample more evenly over the study population. This sampling method was carried out such that the selection process did not interact with any hidden periodic trait within the study population [41]. A systematic sampling interval of 1:22 was used in this study based on Equation (3). Thus, numbers from 1 to 22 were written on pieces of paper and shuffled in a container, one number was selected randomly to determine the penultimate household [42]. After selecting the first household, a spacing of 22 households was observed before selecting the next household for data collection.

Sample size determination using Cochran’s formula at 95% confidence level, 5% level of precision and 50% degree of variability [43]:(1)$$n=\frac{z^{2}p\left(1-\;p\right)}{d^{2}}$$

Proportional allocation:(2)$$n_{h}=\frac{N_{h}}{N}\times n$$

Systematic sample interval (*k*) was estimated as follows [44,45]:(3)$$k=\frac{n}{N}$$
where; n = sample size, z = the confidence interval, p = degree of variability, d = margin of error, N = total number of households, n_h_ = sample size of a community, N_h_ = number of households in a community.

### 2.3. Data Analysis

In this paper, descriptive analysis of data obtained from the survey was done using SPSS (version 26, IBM, Armonk, NY, USA). We assessed relationships including the following: how often households shopped for food; frequency of households’ food cooking at home; the relationship between how often households shopped for food and the storage period of selected common food commodities, including cassava, plantain, maize, rice, tomato and pepper; the relationship between how often food was cooked at home and the storage period of cassava, plantain, maize, rice, tomatoes and peppers. These are some of the major food commodities used for preparing some of the major dishes in Ghana [46]. We evaluated the perception of respondents towards the effect of household-level food storage on food security, signified by whether households always had enough food stored at home, always had easy access to food when stored at home and properly utilized food stored at home. Analyzing statistical association measures was done using Chi-square and Kendall’s tau-b tests. Partial least squares–structural equation modeling (PLS-SEM) was also employed in the analysis of data using SmartPLS software 3.2 [47]. The PLS–SEM approach follows a two-stage analysis: (i) validating the measurement model with reliability and validity tests, and (ii) assessing the structural model with path coefficients, explanatory powers and significance levels. We assessed the reliability and validity of reflective and formative constructs in structural equation modeling [48,49]. The proposed model or framework in this study does not contain any formative constructs. All concepts in the framework were modeled as reflective constructs [50,51]. For this reason, composite reliability and Cronbach’s alpha tests were used to assess reliability.

## 3. Results

### 3.1. Background Characteristics

About 85% of the respondents were females and 15% were males (Table 1). This is attributed to the fact that in Ghana, the responsibilities of planning, shopping and cooking food at home are mostly done by women [52]. A majority of the respondents (40.8%) were between ages 40 and 59 years, and 31.5% had junior high school (JHS) or middle education. Household size mainly (41%) ranged between two to three members and the occupation of most respondents (60.8%) was trading.

### 3.2. How Often Households Shop for Food

The results showed that 66.7% of higher-income households, and 57.7% of upper-middle-income households, in the study areas often shopped for food once a week, while 50%, 66.7% and 47.6% of middle-income, lower-middle-income and lower-income households, respectively shopped for food 2–3 times a week (Figure 1). Additionally, households that shopped for food 4–6 times a week were of upper-middle-income (23.1%), middle-income (6.9%), lower-middle-income (16.7%) and lower-income (15.9%) status. Only 2.2% and 3.1% of lower-income households shopped for food daily and once a month, or less, respectively. Overall, most households shopped for food 2–3 times a week (49%), followed by once a week (24.8%) and 4–6 times a week (14.5%).

### 3.3. Relationship between Food Shopping and Household-Level Food Storage

According to the results, households that often shopped for food once a week mostly stored cassava for 1–3 days (30.3%), plantain for 4–6 days (27.3%), rice for 1–2 weeks (38.4%), maize for 4–6 days (18.2%) and 1–2 weeks (18.2%), tomatoes for 4–6 days (41.4%) and peppers for 4–6 days (51.5%) (Table 2, Table 3 and Table 4). Those that usually shopped for food 2–3 times a week also mostly stored cassava for 1–3 days (49%), plantain for 1–3 days (43.9%), maize for 4–6 days (31.1%), rice for 1–2 weeks (30.6%), tomatoes for 1–3 days (50.5%) and peppers for 1–3 days (44.9%). Additionally, households that often shopped for food 4–6 times a week usually stored cassava, plantain, tomatoes and peppers for 1–3 days (37.9%, 44.8%, 55.2% and 44.8%, respectively) while maize and rice were mostly stored for 4–6 days (31%) and 1–2 weeks (25.9%), respectively. A Chi-square test showed significant association between how often households shopped for food and the storage period of various food commodities (*p* < 0.001). Kendall’s tau-b test also indicated a negative significant association (*p* < 0.001) for all the food commodities, except for cassava and rice. This showed that, as the frequency of food shopping decreased, the storage periods for cassava, plantain, maize, rice, tomatoes and peppers also increased, and decreased with increasing frequency of shopping.

### 3.4. Frequency of Households’ Food Cooking at Home

We found that most middle-income (58.3%), upper-middle-income (61.5%) and higher-income (77.8%) households usually cooked food at home once a day, while those of the lower-income (41.4%) and lower-middle income (47%) status mostly cooked food at home more than once a week (Figure 2). Some higher-income (22.2%), upper-middle-income (34.6%), middle-income (20.8%) and lower-middle-income (7.6%) households also cooked food twice a day. Only 18.5% and 1.5% of lower-income and lower-middle-income households, respectively, cooked less than once a week at home. Households that cooked food at home once a week were of the lower-income (29.5%), lower-middle-income (13.6%) and upper-middle-income (3.8%) statuses. In total, the majority of households often cooked food at home more than once a week (34.8%), followed by once a day (26.3%), and once a week (19.3%).

### 3.5. Relationship between Cooking and Household-Level Food Storage

The relationship between how often food was cooked at home and the storage period of selected food commodities was assessed (Table 5, Table 6 and Table 7). The results revealed that households in the study area that cooked food at home once a day usually stored cassava for 1–3 days (38.1%), plantain for 4–6 days (33.3%), maize for 4–6 days (23.8%), rice for 1–2 weeks (35.2%), tomatoes for 4–6 days (45.7%) and peppers for 4–6 days (47.6%). Those that cooked food at home more than once a week mostly stored cassava for 1–3 days (47.1%), plantain for 1–3 days (51.8%), maize for 4–6 days (31.7%), rice for 1–2 weeks (31.7%), tomatoes for 1–3 days (58.3%), peppers for 1–3 days (46%) and 4–6 days (46%). Additionally, cassava, plantain, tomatoes and peppers were mostly stored for 1–3 days (37.7%, 35.1%, 45.5% and 39%, respectively), maize for 4–6 days (24.7%) and rice for 1–2 weeks (27.3%) by households that cooked food at home once a week. Chi-square and Kendall’s tau-b tests showed significant positive associations (*p* < 0.001) between how often households cooked food at home and the storage period of cassava, plantain, rice, maize, tomatoes and peppers. Therefore, as the frequency of cooking at home decreased, the storage periods for these food commodities decreased, and increased with increasing frequency of cooking.

### 3.6. Respondents’ Perception of Food Storage and Its Effect on Household Food Security

The results revealed that the majority of households agreed *(n* = 186; 46.5%), and some strongly agreed (n = 61; 15.3%), that they always had enough food stored at home (Table 8). However, most of the lower-income households (*n* = 89; 39.2%) disagreed to always having enough food stored at home. A significant association (*p* < 0.001) indicated that as the socio-economic status of households increased, their choice of always having enough food stored at home increased, and decreased with decreasing socio-economic status. Most households also agreed (*n* = 260; 65%), and strongly agreed (*n* = 124; 31%), to having easy access to food when stored at home. There was a positive association between variables, although not significant. A majority of households agreed (*n* = 223; 55.8%), and strongly agreed (*n* = 153; 38.3%), that they properly utilized stored food. A significant negative association (*p* < 0.05) between variables indicated that, as the socio-economic status of households increased from lower- to higher-income status, the opinion of households on properly utilizing food when stored at home also reduced. This showed that the lower the socio-economic status of households, the more they properly utilized food, and vice versa. The findings revealed that household-level food storage contributed to improving household food security.

### 3.7. Structural Model Validation

The concepts of determinants of food storage (knowledge of food storage practices, feeding and cooking behaviors, food choice motives, and food purchasing behavior), food storage practices (food infrastructure, food storage, food handling practices), and food storage outcomes (food waste management, food safety and health, income spent on food and perceived food security) were modeled as multi-item reflective constructs. Composite reliability (CR) values ranged from 0.71 to 1.00, well above the recommended threshold of 0.70 [56]. The average variance extracted (AVE) values also ranged from 0.56 to 1.00, showing acceptable convergent validity of measures, since they were above the recommended 0.50 value [56] (Appendix A). The Heterotrait–Monotriat (HTMT) criterion and Fornell–Larcker criterion were employed to assess discriminant validity [57]. The constructs within the HTMT criterion did not exhibit any discriminant validity issues (Table 9). Regarding the Fornell–Larcker criterion, the square root of the AVE for each factor (diagonal values) was found to be higher than the pair-wise correlation between factors (off-diagonal values) (Table 10). The cross loadings showed items load higher on their respective constructs than on another construct (Appendix A). No item was deleted since they all showed high loadings of above 0.60.

### 3.8. Structural Model Assessment Using Partial Least Squares

The PLS model comprised drivers and inhibitors of household-level food storage behavior and practices, as well as the outcomes of household-level food storage on food wastage, food safety, food expenditure and perceived food security. The PLS algorithm. followed by the standard bootstrapping procedure in SmartPLS, were used to assess hypothesized path coefficients (β), explanatory powers (R^2^) and significance. In terms of drivers and inhibitors of food storage practices among households, the results in Figure 3 showed that two factors (i.e., households’ knowledge of food storage and food shopping behavior) both drove food storage practices among households in the study area. The results showed that households’ knowledge of food storage was positively related to household-level food storage practices. This suggested that, indeed, knowledge of food storage (β = 0.268, *p* value = 0.000, *p* ≤ 0.001) was moderately positive and significantly correlated with household-level food storage practices. Similarly, the results revealed that food shopping behavior drove household-level food storage. However, whilst the relationship between food shopping behavior (β = 0.088, *p* value = 0.006, *p* ≤ 0.01) was positive and significant, the effect was low.

According to the model, factors (i.e., feeding and cooking behaviors and food choice motives) impeded food storage practices among households. The results showed that feeding and cooking behaviors of households impeded household-level food storage practices (Figure 3). In other words, feeding and cooking behaviors (β = −0.005, *p* value = 0.851, n.s.) had a negative effect on household-level food storage. but this effect was low and statistically non-significant (n.s.). This implied that the rampant cooking behavior of households. due to lack of financial resources not favoring cooking in large quantities. but rather in a piece-meal manner, did not encourage food storage at the household level. Similarly, the model revealed that food choice motives impeded household-level food storage among households. The results showed food choice motives (β = −0.356, *p* value = 0.000, *p* ≤ 0.001) to have a largely negative effect on food storage practices of the surveyed households, thereby implying that the food choices made by households did not promote, but rather inhibited, their ability to store food at the household level.

The model also assessed the relationship between knowledge, food handling and household-level food storage. In this regard, we found that households’ knowledge of food storage was positively related to household-level food handling behavior and practices. The results showed that, indeed, knowledge of food storage (β = 0.625, *p* value = 0.000, *p* ≤ 0.001) was highly positive and significantly related to appropriate food handling behavior of households. Additionally, the results found that food handling behaviors positively correlated with household-level food storage practices. Thus, the behavior of households regarding food handling (β = 0.308, *p* value = 0.000, *p* ≤ 0.001) had a moderately positive effect on how they stored food. Furthermore, the findings showed that food choice motives influenced food infrastructural choices of households. Therefore, motives that drove household’s choice of food, such as price fairness, affordability, or convenience, underpinned whether the household would patronize a corner store, farm gate or open market. However, the results revealed a rather moderately negative and significant effect between food shopping (β = −0.349, *p* value = 0.000, *p* ≤ 0.001) and food infrastructure. Similarly, food infrastructure (β = −0.045, *p* value = 0.163, n.s.) exhibited a negatively low but non-significant effect on food storage. This implied that the choice or availability of food infrastructure had a rather detrimental effect on food storage practices of households.

The outcomes of food storage on household food waste management, food safety, expenditure on food and food security was also examined. The model results showed a positively significant relationship between household food storage and food waste management (β = 0.527, *p* value = 0.000, *p* ≤ 0.001), and food safety and healthy attitudes (β = 0.504, *p* value = 0.000, *p* ≤ 0.001) of households in the study communities (Figure 3 and Appendix A). However, the results suggested that household-level food storage practices exhibited a negative effect on food expenditure (β = −0.559, *p* value = 0.000, *p* ≤ 0.001), which implied reduced expense by food-storing households. Contrary to our assumption, that household-level food storage practices had positive impact on perceived food security, the results of the model exhibited a rather negative effect (β = −0.657, *p* value = 0.000, *p* ≤ 0.001).

The results revealed that knowledge of food storage alone explained about 39% (R^2^ = 0.390) of the variance in food handling behavior of households (Figure 4). This implied that ensuring households had adequate knowledge in food storage practices could improve household-level food storage by 39%. Similarly, about 75% of the variance in household-level food storage (R^2^ = 0.754) was explained by the collective effects of food storage knowledge, cooking behaviors, shopping behaviors, food choice motives, food handling and food infrastructure. This implied that the model adequately predicted household-level food storage practices, based on the antecedents and drivers. Hence, effectively managing the knowledge of households about food storage practices, and helping households adjust their cooking patterns and shopping behaviors, as well as their food infrastructure, could highly improve food storage practices among households. In terms of outcomes, household-level food storage accounted for 43% of perceived food security (R^2^ = 0.431), 31% of households’ food expenditure (R^2^ = 0.312), 27% of households’ food waste management behaviors (R^2^ = 0.278) and 25% of food safety practices (R^2^ = 0.254) among the surveyed households in Accra, Ghana.

## 4. Discussion

The study revealed that households’ knowledge of food storage and food shopping behavior were both drivers of household-level food storage. Knowledge is critical with regards to improving food storage at the household level. When most members of households or even the food head of household is able to acquire some training in food storage, it helps to improve their knowledge to understand and properly handle different food types to promote shelf life, and prevent food from being damaged quickly, thereby avoiding wastage. Our findings supported that of a study by [58], which indicated that respondents had moderate knowledge in food storage (64.9%) and, though they had general knowledge in food storage, they lacked detailed understanding and this had an adverse effect on their food storage practices. The findings of [59] showed that the level of knowledge of food vendors rose from an average of 24.35% to 66.2% after training intervention. According to the work of [60], trained food handlers had significantly advanced knowledge relative to untrained food handlers, which suggested a need for training. This agrees with the findings of this study that the knowledge of sampled households on food storage positively correlated with food handling practices and could improve household-level food storage by 39%.

Food shopping behavior had a significantly positive relationship with household-level food storage according to the model. Nevertheless, the survey results showed a significantly negative effect between food shopping and the period food commodities remained in storage within households, with the exceptions of cassava and rice. The frequency of food shopping differed among households depending on individual transportation options, time for shopping, availability and accessibility of food infrastructures, and availability of proper storage facilities. The frequency of shopping trips had an influence on the amount of food that remained uneaten in homes [61], and, hence, the amount of food stored at home and the period of storage. High frequency of shopping trips did not make it necessary to store food at home, especially perishable foods [62]. Nevertheless, households in the study area continued to store food as they shopped to ensure they always had enough food stored at home to serve as a buffer during critical times.

There was a moderately negative and significant effect between food shopping and food infrastructure. Households chose to purchase food from the usual food sources for the following reasons: low food prices; bulk purchase deals; bargaining power; and being able to obtain fresh, good quality and different varieties of food items. A study by [63] reported that the food retail system in Ghana is dominated by open-air markets and that 70% of households usually shop for food from open-air markets (at least once a week). The findings agree with a study by [64], which showed that households did not shop at stores near them, but travelled 3.8 miles to their usual stores to buy food, because of affordability of food prices, good quality and diversity of available foods to households. Additionally, the findings from the survey revealed that, as the frequency of cooking plantain, cassava, maize, rice, pepper and tomato decreased, the storage periods for these foods decreased, and increased with increasing frequency of cooking. The model also showed that cooking had a low and non-significant negative effect on household-level food storage, hence, frequent feeding and cooking might not encourage food storage. A study by [65] reported that participants who cooked and ate at home more than five times per week consumed 62.3 g and 97.8 g more fruits and vegetables, respectively, than those who cooked less than three times per week, and, hence, the more consumers cooked and ate at home, the less food commodities remained in storage.

The study also found that household-level food storage had a significant positive effect on food waste management. This meant that households were able to reduce food wastage as a result of storing food at home. According to the 2021 Food Waste Index Report, approximately 931 million tons of food waste was produced in 2019, with 61% coming from households [66]. A research conducted by [67] to assess the extent to which 120 households from the Accra Metropolitan Area in Ghana contributed to food wastage found that the food waste generation rate of households was, on average, 0.12 kg/person/day. Reducing food wastage, especially within households, is key and should be promoted through appropriate technologies, including improved food storage techniques. Minimizing food loss or wastage by 50% by the year 2025 in Ghana would minimize the cost of food production, increase food production, enhance revenue generation, and improve food security [68].

The findings showed that household-level food storage had a significant positive effect on food safety and health. Food is regarded to be safe when it is free from hazards that may pose risk or threat to the health of consumers [69]. Unsafe foods are usually contaminated by microbes, which render them unfit for consumption and make consumers susceptible to foodborne illness, hence, making food safety an important public health concern. A study by [70] in Accra assessing public concerns and perception about food safety found that between 50% and 90% of the consuming public was either extremely, or very, worried with regards to all the risks and hazards associated with food safety. In order to minimize the risk or threat of eating unsafe foods, consumers must be ready to change attitudes and behaviors which are inconsistent with safe food storage and handling practices [18].

Our results revealed that household-level food storage had a significant negative effect on household food expenditure. This implied that households in the study communities were able to reduce the amount of income spent on food when they stored food at home. The findings supported an assertion by [71] that, by taking advantage of abundant food supply, bulk purchase deals and low food prices, particularly during glut seasons, households save considerable amounts of money when they buy food to store at home for future use. This could explain the reduction in food expenditure resulting from household-level food storage among the sampled households. The model also showed that household-level food storage had a negative effect on perceived food security, although it could contribute to improving food security by 43%. This implied that, whilst household-level food storage helped the sampled households to effectively manage food waste, ensure food safety and reduce food expenditure, it might not necessarily yield a positive outlook regarding completely achieving food security. According to the perception of most respondents, their households always had enough food stored at home, had easy access to food when stored at home and properly utilized stored food. Achieving food security involves properly storing food and consistently making adequate amounts of nutritious foods available and accessible [72]. When food is always available, adequately accessible, properly utilized, with no risks affecting them, then food security is said to be achieved [8]. Although households have available and accessible food when it is stored at home, properly utilizing the stored food is critical to meet their dietary needs [25]. The ability of most households in the studied communities to continually obtain enough stored nutritious foods might be a challenge. Recent findings by [36], assessing household dietary diversity in Accra, revealed low intake of foods that are rich in micronutrients by households, although there was high dietary diversity. Ensuring the achievement of food security, particularly within households, does not, therefore, necessarily depend exclusively on food storage, although food storage is necessary.

## 5. Conclusions

Understanding how consumer behavior affects food storage at the household level can aid in the improvement of food safety and security for households in urban areas. This study’s findings showed that knowledge of food storage was positively and significantly related to food storage. Household-level food storage minimizes food wastage, enhances food safety, reduces food expenditure and could contribute to helping achieve household food security, according to the findings. There was a negative correlation between the frequency with which households go grocery shopping and time food was kept in storage, although food shopping was a driver of food storage within households. Cooking inhibited food storage, but the frequency with which cooking was done by households had a substantial positive association with the storage time of food commodities. Most respondents indicated that they always had enough food stored, had easily accessible stored food and made good use of food stored at home. The frequency and ability of households to make food choices, including to shop, cook, and store food, were among key determinants when assessing the impact of household-level food storage on food safety and security.

Households lacking access to adequate funds for food purchases and storage may be at a higher risk of food insecurity. However, low-income households require long-term solutions that improve both their physical and financial access to food. For instance, regulations should be geared at supporting periodic market days, such as once a week, when fresh, safe, good quality, and inexpensive food commodities would be promoted in cities, where people often shop at informal food sources, such as open-air markets, wet markets, and farmer markets. To guarantee that low-income families have access to affordable local food, food price policies and social interventions may include setting a maximum price for food items. Furthermore, it is important to advocate for conditional food and monetary allocations, which play a crucial role in reducing socio-economic inequities. Modernizing or upgrading conventional home-level food preservation, and storage practices that are inexpensive and easily implemented, would enable households, especially low-income households, to properly store more food and be assured of household food security.

## Figures and Tables

**Figure 1 foods-11-03266-f001:**
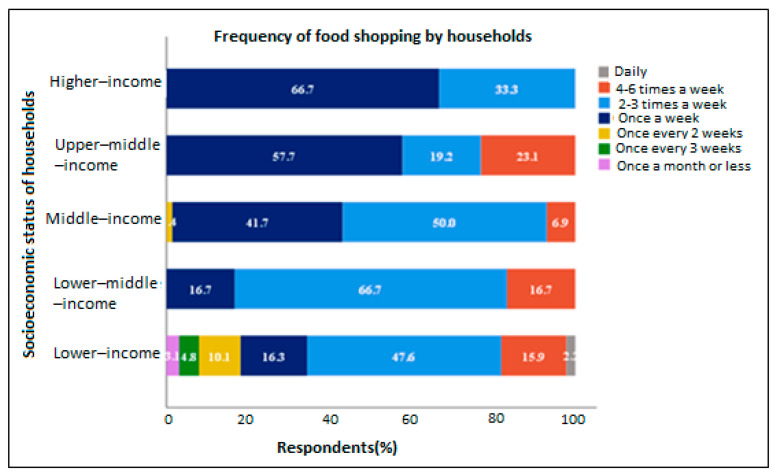
How often households shopped for food.

**Figure 2 foods-11-03266-f002:**
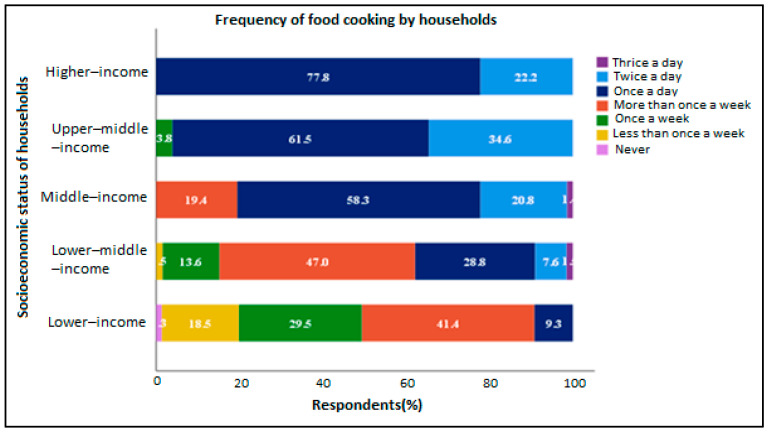
How often households cooked food at home.

**Figure 3 foods-11-03266-f003:**
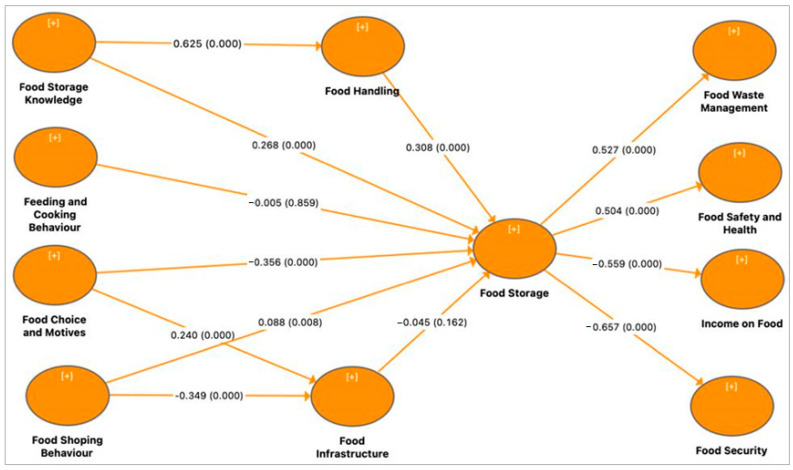
Network analysis of determinants of household-level food storage practices and outcomes on food safety and security. The +/- indicates the strength of the relationship between and among the clustering variables. The [+] indicates positive relationship between clustering variables at *p* ≤ 0.05 level of significance.

**Figure 4 foods-11-03266-f004:**
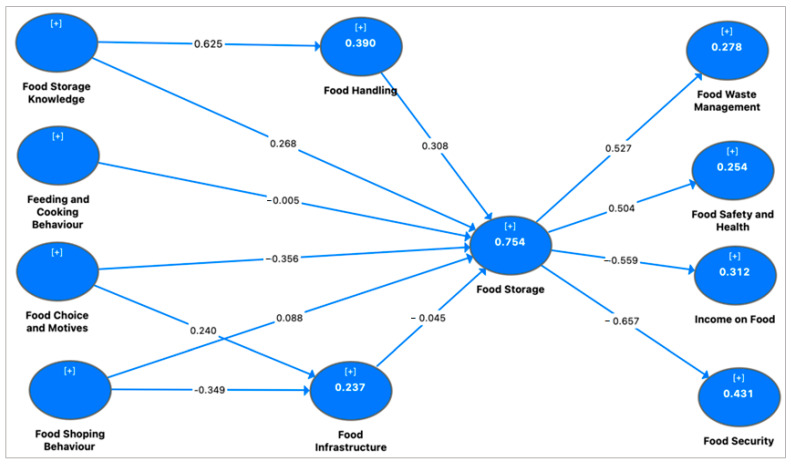
Network analysis of determinants of household-level food storage practices and outcomes on food safety and security. The +/- indicates the strength of the relationship between and among the clustering variables. The [+] indicates positive relationship between clustering variables.

**Table 1 foods-11-03266-t001:** Socio-demographic distribution of respondents and households.

Variable	Percentage	Variable	Percentage
*Age of respondent*		*Household size*	
Less than 18 years	0.3	1	18.8
18–25 years	13.0	2–3	41.0
26–39 years	38.8	4–5	28.5
40–59 years	40.8	6 or more	11.8
60 years and above	7.2	*Socio-economic status of household (SES)*	
*Sex of respondent*		Lower-income /poorest	56.8
Male	15.0	Lower-middle-income	16.5
Female	85.0	Middle-income	18.0
*Education of respondent*		Upper-middle-income	6.5
None	7.5	Higher-income/richest	2.3
Primary	19.5	*Education of household head*	
JHS/middle	31.5	None	8.0
SHS/secondary	25.8	Primary	17.8
Tertiary (Degree/diploma)	15.8	JHS/middle	28.0
*Occupation of respondent*		SHS/secondary	26.0
None	1.0	Tertiary (Degree/diploma)	20.3
Professional/technical/managerial/clerical	6.3	*Occupation of household head*	
Agricultural self-employed	0.8	None	0.0
Trade	60.8	Professional/technical/managerial/clerical	13.0
Service	6.3	Agricultural self-employed	2.5
Skilled manual	15.8	Trade	41.8
Unskilled manual	9.3	Service	7.2
*Monthly income of respondent (GH* *Ȼ* *)*		Skilled manual	23.0
500 and below	11.3	Unskilled manual	12.5
501–1000	38.0	*Monthly income of household head (GH* *Ȼ* *)*	
1001–1500	15.5	500 and below	6.8
1501–2000	2.0	501–1000	39.5
2001–2500	10.5	1001–1500	13.8
above 2500	22.8	1501–2000	4.0
		2001–2500	7.0
		above 2500	29.0

Total sample size was 400 for all variables. Socio-economic status of households was estimated using principal component analysis (PCA) [53,54,55]. Junior High School—JHS, Senior High School—SHS, Ghana Cedi—GHȻ.

**Table 2 foods-11-03266-t002:** Relationship between frequency of household food shopping and storage of cassava and plantain.

Frequency of Household Food Shopping	Storage Period for Cassava					
	3–4 Weeks (*n* = 6)	1–2 Weeks (*n* = 19)	4–6 Days (*n* = 60)	1–3 Days (*n* = 151)	Never (*n* = 164)	Total (*n* = 400)
Daily	0(0%)	0(0%)	0(0%)	0(0%)	5(100%)	5(100%)
4–6 times a week	0(0%)	5(8.6%)	8(13.8%)	22(37.9%)	23(39.7%)	58(100%)
2–3 times a week	4(2%)	4(2%)	26(13.3%)	96(49%)	66(33.7%)	196(100%)
Once a week	2(2%)	6(6.1%)	17(17.2%)	30(30.3%)	44(44.4%)	99(100%)
Once every 2 weeks	0(0%)	2(8.3%)	3(12.5%)	2(8.3%)	17(70.8%)	24(100%)
Once every 3 weeks	0(0%)	0(0%)	4(36.4%)	0(0%)	7(63.6%)	11(100%)
Once a month or less	0(0%)	2(28.6%)	2(28.6%)	1(14.3%)	2(28.6%)	7(100%)
*Chi-square = 63.074 (p ≤ 0.000); Kendall’s tau-b = −0.024 (p ≤ 0.623)*						
**Frequency of Household Food Shopping**	**Storage Period for Plantain**					
	**3–4 Weeks** **(*n* = 4)**	**1–2 Weeks** **(*n* = 45)**	**4–6 Days** **(*n* = 102)**	**1–3 Days** **(*n* = 146)**	**Never** **(*n* = 103)**	**Total** **(*n* = 400)**
Daily	0(0%)	0(0%)	1(20%)	4(80%)	0(0%)	5(100%)
4–6 times a week	0(0%)	7(12.1%)	18(31%)	26(44.8%)	7(12.1%)	58(100%)
2–3 times a week	4(2%)	24(12.2%)	55(28.1%)	86(43.9%)	27(13.8%)	196(100%)
Once a week	0(0%)	14(14.1%)	27(27.3%)	23(23.2%)	35(35.4%)	99(100%)
Once every 2 weeks	0(0%)	0(0%)	1(4.2%)	7(29.2%)	16(66.7%)	24(100%)
Once every 3 weeks	0(0%)	0(0%)	0(0%)	0(0%)	11(100%)	11(100%)
Once a month or less	0(0%)	0(0%)	0(0%)	0(0%)	7(100%)	7(100%)
*Chi-square = 118.155 (p ≤ 0.000); Kendall’s tau-b = −0.245 (p ≤ 0.000)*						

**Table 3 foods-11-03266-t003:** Relationship between frequency of household food shopping and storage of maize and rice.

Frequency of Household Food Shopping	Storage Period for Maize						
	More Than 1 Month (*n* = 0)	3–4 Weeks (*n* = 21)	1–2 Weeks (*n* = 49)	4–6 Days (*n* = 100)	1–3 Days (*n* = 74)	Never (*n* = 156)	Total (*n* = 400)
Daily	0(0%)	0(0%)	0(0%)	1(20%)	0(0%)	4(80%)	5(100%)
4–6 times a week	0(0%)	4(6.9%)	4(6.9%)	18(31%)	16(27.6%)	16(27.6%)	58(100%)
2–3 times a week	0(0%)	10(5.1%)	27(13.8%)	61(31.1%)	45(23%)	53(27%)	196(100%)
Once a week	0(0%)	7(7.1%)	18(18.2%)	18(18.2%)	10(10.1%)	46(46.5%)	99(100%)
Once every 2 weeks	0(0%)	0(0%)	0(0%)	2(8.3%)	3(12.5%)	19(79.2%)	24(100%)
Once every 3 weeks	0(0%)	0(0%)	0(0%)	0(0%)	0(0%)	11(100%)	11(100%)
Once a month or less	0(0%)	0(0%)	0(0%)	0(0%)	0(0%)	7(100%)	7(100%)
*Chi-square = 82.195 (p ≤ 0.000); Kendall’s tau-b = −0.174 (p ≤ 0.000)*							
**Frequency of Household Food Shopping**	**Storage Period for Rice**						
	**More Than 1 Month** **(*n* = 21)**	**3–4 Weeks** **(*n* = 49)**	**1–2 Weeks** **(*n* = 119)**	**4–6 Days** **(*n* = 64)**	**1–3 Days** **(*n* = 70)**	**Never** **(*n* = 77)**	**Total** **(*n* = 400)**
Daily	0(0%)	0(0%)	0(0%)	1(20%)	3(60%)	1(20%)	5(100%)
4–6 times a week	4(6.9%)	5(8.6%)	15(25.9%)	6(10.3%)	12(20.7%)	16(27.6%)	58(100%)
2–3 times a week	8(4.1%)	29(14.8%)	60(30.6%)	40(20.4%)	43(21.9%)	16(8.2%)	196(100%)
Once a week	9(9.1%)	14(14.1%)	38(38.4%)	10(10.1%)	8(8.1%)	20(20.2)	99(100%)
Once every 2 weeks	0(0%)	1(4.2%)	5(20.8)	3(12.5%)	3(12.5%)	12(50%)	24(100%)
Once every 3 weeks	0(0%)	0(0%)	1(9.1%)	4(36.4%)	1(9.1%)	5(45.5%)	11(100%)
Once a month or less	0(0%)	0(0%)	0(0%)	0(0%)	0(0%)	7(100%)	7(100%)
*Chi-square = 99.434 (p ≤ 0.000); Kendall’s tau-b = −0.039 (p ≤ 0.394)*							

**Table 4 foods-11-03266-t004:** Relationship between frequency of household food shopping and storage of tomato and pepper.

Frequency of Household Food Shopping	Storage Period for Tomatoes					
	3–4 Weeks (*n* = 22)	1–2 Weeks (*n* = 16)	4–6 Days (*n* = 139)	1–3 Days (*n* = 168)	Never (*n* = 55)	Total (*n* = 400)
Daily	0(0%)	0(0%)	0(0%)	5(100%)	0(0%)	5(100%)
4–6 times a week	5(8.6%)	3(5.2%)	16(27.6%)	32(55.2%)	2(3.4%)	58(100%)
2–3 times a week	12(6.1%)	5(2.6%)	77(39.3%)	99(50.5%)	3(1.5%)	196(100%)
Once a week	5(5.1)	8(8.1%)	41(41.4%)	24(24.2%)	21(21.2%)	99(100%)
Once every 2 weeks	0(0%)	0(0%)	4(16.7%)	6(25%)	14(58.3%)	24(100%)
Once every 3 weeks	0(0%)	0(0%)	1(9.1%)	2(18.2%)	8(72.7%)	11(100%)
Once a month or less	0(0%)	0(0%)	0(0%)	0(0%)	7(100%)	7(100%)
*Chi-square = 176.951 (p ≤ 0.000); Kendall’s tau-b = −0.175 (p ≤ 0.000)*						
**Frequency of Household Food Shopping**	**Storage Period for Peppers**					
	**3–4 Weeks** **(*n* = 21)**	**1–2 Weeks** **(*n* = 28)**	**4–6 Days** **(*n* = 158)**	**1–3 Days** **(*n* = 150)**	**Never** **(*n* = 43)**	**Total** **(*n* = 400)**
Daily	0(0%)	0(0%)	0(0%)	5(100%)	0(0%)	5(100%)
4–6 times a week	5(8.6%)	7(12.1%)	19(32.8%)	26(44.8%)	1(1.7%)	58(100%)
2–3 times a week	12(6.1%)	12(6.1%)	83(42.3%)	88(44.9%)	1(0.5%)	196(100%)
Once a week	4(4%)	9(9.1%)	51(51.5%)	18(18.2%)	17(17.2%)	99(100%)
Once every 2 weeks	0(0%)	0(0%)	4(16.7%)	11(45.8%)	9(37.5%)	24(100%)
Once every 3 weeks	0(0%)	0(0%)	1(9.1%)	2(18.2%)	8(72.7%)	11(100%)
Once a month or less	0(0%)	0(0%)	0(0%)	0(0%)	7(100%)	7(100%)
*Chi-square = 183.619 (p ≤ 0.000); Kendall’s tau-b = −0.175 (p ≤ 0.000)*						

**Table 5 foods-11-03266-t005:** Relationship between frequency of cooking at home and storage of cassava and plantain.

Frequency of Food Cooking by Households	Storage Period for Cassava					
	3–4 Weeks (*n* = 6)	1–2 Weeks (*n* = 19)	4–6 Days (*n* = 60)	1–3 Days (*n* = 151)	Never (*n* = 164)	Total (*n* = 400)
Thrice a day	0(0%)	0(0%)	1(50%)	0(0%)	1(50%)	2(100%)
Twice a day	2(6.5%)	3(9.7%)	7(22.6%)	9(29%)	10(32.3%)	31(100%)
Once a day	4(3.8%)	8(7.6%)	24(22.9%)	40(38.1%)	29(27.6%)	105(100%)
More than once a week	0(0%)	1(0.7%)	10(7.2%)	68(48.9%)	60(43.2%)	139(100%)
Once a week	0(0%)	3(3.9%)	8(10.4%)	29(37.7%)	37(48.1%)	77(100%)
Less than once a week	0(0%)	3(7%)	8(18.6%)	5(11.6%)	27(62.8%)	43(100%)
Never	0(0%)	1(33.3%)	2(66.7%)	0(0%)	0(0%)	3(100%)
*Chi-square = 73.720 (p ≤ 0.000); Kendall’s tau-b = 0.180 (p ≤ 0.000)*						
**Frequency of Food Cooking by Households**	**Storage Period for Plantain**					
	**3–4 Weeks** **(*n* = 4)**	**1–2 Weeks** **(*n* = 45)**	**4–6 Days** **(*n* = 102)**	**1–3 Days** **(*n* = 146)**	**Never** **(*n* = 103)**	**Total** **(*n* = 400)**
Thrice a day	0(0%)	0(0%)	2(100%)	0(0%)	0(0%)	2(100%)
Twice a day	1(3.2%)	5(16.1%)	15(48.4%)	8(25.8%)	2(6.5%)	31(100%)
Once a day	3(2.9%)	18(17.1%)	35(33.3%)	33(31.4%)	16(15.2%)	105(100%)
More than once a week	0(0%)	13(9.4%)	28(20.1%)	72(51.8%)	26(18.7%)	139(100%)
Once a week	0(0%)	9(11.7%)	17(22.1%)	27(35.1%)	24(31.2%)	77(100%)
Less than once a week	0(0%)	0(0%)	5(11.6%)	6(14%)	32(74.4%)	43(100%)
Never	0(0%)	0(0%)	0(0%)	0(0%)	3(100%)	3(100%)
*Chi-square = 114.247 (p ≤ 0.000); Kendall’s tau-b = 0.321 (p ≤ 0.000)*						

**Table 6 foods-11-03266-t006:** Relationship between frequency of food cooking by households and storage of maize and rice.

Frequency of Food Cooking by Households	Storage Period for Maize						
	More Than 1 Month (*n* = 0)	3–4 Weeks (*n* = 21)	1–2 Weeks (*n* = 49)	4–6 Days (*n* = 100)	1–3 Days (*n* = 74)	Never (*n* = 156)	Total (*n* = 400)
Thrice a day	0(0%)	0(0%)	1(50%)	0(0%)	0(0%)	1(50%)	2(100%)
Twice a day	0(0%)	4(12.9%)	6(19.4%)	9(29%)	4(12.9%)	8(25.8%)	31(100%)
Once a day	0(0%)	13(12.4%)	21(20%)	25(23.8%)	12(11.4%)	34(32.4%)	105(100%)
More than once a week	0(0%)	3(2.2%)	16(11.5%)	44(31.7%)	38(27.3%)	38(27.3%)	139(100%)
Once a week	0(0%)	1(1.3%)	4(5.2%)	19(24.7%)	16(20.8%)	37(48.1%)	77(100%)
Less than once a week	0(0%)	0(0%)	1(2.3%)	3(7%)	4(9.3%)	35(81.4%)	43(100%)
Never	0(0%)	0(0%)	0(0%)	0(0%)	0(0%)	3(100%)	3(100%)
*Chi-square = 90.410 (p ≤ 0.000); Kendall’s tau-b = 0.291 (p ≤ 0.000)*							
**Frequency of Food Cooking by Households**	**Storage Period for Rice**						
	**More Than 1 Month (*n* = 21)**	**3–4 Weeks** **(*n* = 49)**	**1–2 Weeks** **(*n* = 119)**	**4–6 Days** **(*n* = 64)**	**1–3 Days** **(*n* = 70)**	**Never** **(*n* = 77)**	**Total** **(*n* = 400)**
Thrice a day	0(0%)	1(50%)	1(50%)	0(0%)	0(0%)	0(0%)	2(100%)
Twice a day	4(12.9%)	15(48.4%)	9(29%)	3(9.7%)	0(0%)	0(0%)	31(100%)
Once a day	14(13.3%)	21(20%)	37(35.2%)	11(10.5%)	9(8.6%)	13(12.4%)	105(100%)
More than once a week	3(2.2%)	7(5%)	44(31.7%)	29(20.9%)	40(28.8%)	16(11.5%)	139(100%)
Once a week	0(0%)	4(5.2%)	21(27.3%)	12(15.6%)	16(20.8%)	24(31.2%)	77(100%)
Less than once a week	0(0%)	1(2.3%)	7(16.3%)	9(20.9%)	5(11.6%)	21(48.8%)	43(100%)
Never	0(0%)	0(0%)	0(0%)	0(0%)	0(0%)	3(100%)	3(100%)
*Chi-square = 161.636 (p ≤ 0.000); Kendall’s tau-b = 0.402 (p ≤ 0.000)*							

**Table 7 foods-11-03266-t007:** Relationship frequency of food cooking by households and storage of tomato and pepper.

Frequency of Food Cooking by Households	Storage Period for Tomatoes					
	3–4 Weeks (*n* = 22)	1–2 Weeks (*n* = 16)	4–6 Days (*n* = 139)	1–3 Days (*n* = 168)	Never (*n* = 55)	Total (*n* = 400)
Thrice a day	0(0%)	0(0%)	2(100%)	0(0%)	0(0%)	2(100%)
Twice a day	8(25.8%)	2(6.5%)	14(45.2%)	7(22.6%)	0(0%)	31(100%)
Once a day	14(13.3%)	8(7.6%)	48(45.7%)	34(32.4%)	1(1%)	105(100%)
More than once a week	0(0%)	6(4.3%)	49(35.3%)	81(58.3%)	3(2.2%)	139(100%)
Once a week	0(0%)	0(0%)	20(26%)	35(45.5%)	22(28.6%)	77(100%)
Less than once a week	0(0%)	0(0%)	6(14%)	11(25.6%)	26(60.5%)	43(100%)
Never	0(0%)	0(0%)	0(0%)	0(0%)	3(100%)	3(100%)
*Chi-square = 219.922 (p ≤ 0.000); Kendall’s tau-b = 0.478 (p ≤ 0.000)*						
**Frequency of Food Cooking by Households**	**Storage Period for Peppers**					
	**3–4 Weeks** **(*n* = 21)**	**1–2 Weeks** **(*n* = 28)**	**4–6 Days** **(*n* = 158)**	**1–3 Days** **(*n* = 150)**	**Never** **(*n* = 43)**	**Total** **(*n* = 400)**
Thrice a day	0(0%)	0(0%)	1(50%)	1(50%)	0(0%)	2(100%)
Twice a day	8(25.8%)	3(9.7%)	14(45.2%)	6(19.4%)	0(0%)	31(100%)
Once a day	13(12.4%)	11(10.5%)	50(47.6%)	31(29.5%)	0(0%)	105(100%)
More than once a week	0(0%)	10(7.2%)	64(46%)	64(46%)	1(0.7%)	139(100%)
Once a week	0(0%)	4(5.2%)	25(32.5%)	30(39%)	18(23.4%)	77(100%)
Less than once a week	0(0%)	0(0%)	4(9.3%)	18(41.9%)	21(48.8%)	43(100%)
Never	0(0%)	0(0%)	0(0%)	0(0%)	3(100%)	3(100%)
*Chi-square = 198.678 (p ≤ 0.00); Kendall’s tau-b = 0.43 (p ≤ 0.00)*						

**Table 8 foods-11-03266-t008:** The perception of respondents about food storage effect on household food security.

Sufficient food stored at home by households						
Sufficient Food Stored at Home by Households	Strongly Disagree (*n* = 43)	Disagree (*n* = 91)	Neither Agree Nor Disagree (*n* = 19)	Agree (*n* = 186)	Strongly Agree (*n* = 61)	Total (*n* = 400)
Lower-income	43(18.9%)	89(39.2%)	18(7.9%)	68(30%)	9(4%)	227(100%)
Lower-middle-income	0(0%)	2(3%)	1(1.5%)	49(74.2%)	14(21.2%)	66(100%)
Middle-income	0(0%)	0(0%)	0(0%)	46(63.9%)	26(36.1%)	72(100%)
Upper-middle-income	0(0%)	0(0%)	0(0%)	19(73.1%)	7(26.9%)	26(100%)
Higher-income	0(0%)	0(0%)	0(0%)	4(44.4%)	5(55.6%)	9(100%)
*Chi-square = 192.664 (p ≤ 0.000); Kendall’s tau-b = 0.558 (p ≤ 0.000)*						
**Households’ easy access to food when stored at home**						
**Households’ Easy Access to Food When Stored at Home**	**Strongly Disagree** **(*n* = 0)**	**Disagree** **(*n* = 16)**	**Neither Agree nor Disagree** **(*n* = 0)**	**Agree** **(*n* = 260)**	**Strongly Agree** **(*n* = 124)**	**Total** **(*n* = 400)**
Lower-income	0(0%)	9(4%)	0(0%)	149(65.6%)	69(30.4%)	227(100%)
Lower-middle-income	0(0%)	6(9.1%)	0(0%)	42(63.6%)	18(27.3%)	66(100%)
Middle-income	0(0%)	1(1.4%)	0(0%)	49(68.1%)	22(30.6%)	72(100%)
Upper-middle-income	0(0%)	0(0%)	0(0%)	16(61.5%)	10(38.5%)	26(100%)
Higher-income	0(0%)	0(0%)	0(0%)	4(44.4%)	5(55.6%)	9(100%)
*Chi-square = 10.218 (p ≤ 0.250); Kendall’s tau-b = 0.041 (p ≤ 0.373)*						
**Food utilization by households when stored at home**						
**Proper Household Food Utilization When Stored at Home**	**Strongly Disagree** **(*n* = 0)**	**Disagree** **(*n* = 19)**	**Neither Agree nor Disagree** **(*n* = 5)**	**Agree** **(*n* = 223)**	**Strongly Agree** **(*n* = 153)**	**Total** **(*n* = 400)**
Lower-income	0(0%)	12(5.3%)	2(0.9%)	110(48.5%)	103(45.4%)	227(100%)
Lower-middle-income	0(0%)	6(9.1%)	2(3%)	44(66.7%)	14(21.2%)	66(100%)
Middle-income	0(0%)	0(0%)	1(1.4%)	50(69.4%)	21(29.2%)	72(100%)
Upper-middle-income	0(0%)	1(3.8%)	0(0%)	15(57.7%)	10(38.5%)	26(100%)
Higher-income	0(0%)	0(0%)	0(0%)	3(33.3%)	6(66.7%)	9(100%)
*Chi-square = 25.518 (p ≤ 0.013); Kendall’s tau-b = −0.093 (p ≤ 0.038)*						

**Table 9 foods-11-03266-t009:** Relationship between food storage determinants, practices and outcomes (Heterotrait–Monotriat Ratio).

	1	2	3	4	5	6	7	8	9	10	11	12	13	14
1. Feeding-Cooking Behavior														
2. Food Choice Motives	0.478													
3. Food Handling	0.490	0.728												
4. Food Infrastructure	0.335	0.812	0.616											
5. Food Safety and Health	0.338	0.596	0.666	0.907										
6. Food Security	0.410	0.872	0.770	0.909	0.552									
7. Food Shopping Behavior	0.420	0.596	0.795	0.807	0.655	0.576								
8. Food Storage	0.415	0.881	0.811	0.862	0.771	0.804	0.889							
9. Food Storage Knowledge	0.445	0.810	0.440	0.811	0.883	0.750	0.694	0.992						
10. Food Waste Management	0.438	0.825	0.858	0.976	0.647	0.725	0.775	0.809	0.797					
11. Household Head Income	0.432	0.957	0.666	0.906	0.537	0.829	0.520	0.782	0.802	0.675				
12. Household Size	0.194	0.281	0.273	0.658	0.338	0.364	0.490	0.335	0.497	0.345	0.233			
13. Socio-economic Status	0.490	0.929	0.657	0.794	0.566	0.826	0.571	0.812	0.760	0.787	0.861	0.342		
14. Income on Food	0.355	0.716	0.522	0.490	0.392	0.445	0.375	0.611	0.478	0.558	0.698	0.100	0.697	

**Table 10 foods-11-03266-t010:** Relationship between determinants of food storage, practices and outcomes (Fornell–Larcker Criterion).

	1	2	3	4	5	6	7	8	9	10	11	12	13	14
1. Feeding-Cooking Behavior	1.000													
2. Food Choice and Motives	0.427	0.910												
3. Food Handling	−0.318	−0.507	0.762											
4. Food Infrastructure	0.142	0.360	−0.528	0.750										
5. Food Safety and Health	−0.242	−0.408	0.525	−0.344	0.804									
6. Food Security	0.360	0.668	−0.430	0.307	−0.353	0.765								
7. Food Shopping Behavior	−0.260	−0.345	0.501	−0.431	0.446	-0.279	0.777							
8. Food Storage	−0.381	−0.721	0.725	−0.537	0.503	−0.615	0.528	0.769						
9. Food Storage Knowledge	−0.359	−0.597	0.625	−0.610	0.498	−0.473	0.530	0.749	0.856					
10. Food Waste Management	−0.309	−0.523	0.446	−0.376	0.498	−0.412	0.344	0.527	0.465	0.813				
11. Household Head Income	−0.432	−0.851	0.512	−0.445	0.409	−0.712	0.341	0.721	0.661	0.478	1.000			
12. Household Size	−0.194	−0.251	0.222	−0.394	0.233	0.022	0.284	0.306	0.394	0.256	0.233	1.000		
13. Socio-economic Status	−0.490	−0.899	0.511	−0.395	0.425	−0.698	0.358	0.744	0.629	0.562	0.861	0.342	1.000	
14. Income on Food	0.352	0.632	−0.360	0.209	−0.300	0.924	−0.235	−0.559	−0.397	−0.392	−0.692	0.096	−0.691	0.956

## Data Availability

Data is contained within the article.

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
