# Peer review of "Determinants of Household-Level Food Storage Practices and Outcomes on Food Safety and Security in Accra, Ghana"

_foods, 2022, doi:10.3390/foods11203266_

Round 1
Reviewer 1 Report
Comment: Thank you for the opportunity to review the manuscript entitled “Determinants of Household-Level Food Storage Practices and Outcomes on Food Safety and Security”, which aims to: (a) assess the determinants of household-level food storage; (b) ascertain consumers' behavior and perception towards food storage; and (c) assess the effect of household-level food storage on food safety, wastage, food expenditure and security. The novelty of the research is the areas of the research, namely Dzorwulu and Jamestown, both located in Accra, Ghana. The research adopts a survey method to collect data, and a structural equation modelling to evaluate results. The research is interesting, original and well-conducted. I have some minor/major concerns related to the study, but overall my feedback is very positive.
Title: Considering that the research investigates Dzorwulu and Jamestown, both located in Accra, Ghana, I would include the geographical areas in the title, as follows: “Determinants of Household-Level Food Storage Practices and Outcomes on Food Safety and Security in Accra, Ghana” or solely “Determinants of Household-Level Food Storage Practices and Outcomes on Food Safety and Security in Ghana”. I believe it can enhance the interest among readers.
Abstract: The abstract is clear and comprehensive. The authors have included the (brief) theoretical background of the research, the aims and scope, the methods applied and the main results of the research. Further, the authors have included some highlights. The abstract accounts for 245 words.
Introduction: “Introduction” is clear and comprehensive, but some minor concerns must be addressed to enhance its clarity. In lines 34-42 I would include some more references, specific to the different purposes of “storing food”. It can highlight that the main issues raised by your research have been already investigated in the scientific literature.
Lines 43-47 provide the definition of “food security”. Considering that “food security” and “food safety” are different concept, could the authors please provide some information related to “food safety”? Further, when discussion the previous research on the topic, please highlight that food storage is closely related to food preservation and human health.
Lines74-80. Is it possible to cite more recent and updated references related to consumer behavior and food storage at household? References [19-34] are not recent. For instance:
Amicarelli, V., Lagioia, G., Sampietro, S. and Bux, C. (2022). Has the COVID-19 pandemic changed food waste perception and behavior? Evidence from Italian consumers. Socio-Economic Planning Sciences, 82, Part A, 101095. 10.1016/j.seps.2021.101095
Alvarez de los Mozos, E., Badurdeen, F., Dossou, P-E. (2020). Sustainable Consumption by Reducing Food Waste: A Review of the Current State and Directions for Future Research. Procedia Manufacturing, 51, 1791-1798. https://doi.org/10.1016/j.promfg.2020.10.249.
van Holsteijn, F., Kemna, R. (2018). Minimizing food waste by improving storage conditions in household refrigeration. Resources, Conservation and Recycling, 128, 25-31,https://doi.org/10.1016/j.resconrec.2017.09.012.
Goossens, Y., Wegner, A., Schmidt, T. (2019). Sustainability Assessment of Food Waste Prevention Measures: Review of Existing Evaluation Practices. Front. Sustain. Food Syst. 3:90. 10.3389/fsufs.2019.00090
A paragraph should be devoted to the reasons why the authors investigate food security and food safety in Ghana. The authors only say that “In this paper, we assessed the determinants of household-level food storage practices and outcomes on food safety and security among households in Accra, Ghana” (lines 80-82), but nothing is said to justify “why” the authors have decided to investigate such an area. Is it possible to provide some data related to food security in Ghana, as well as food security and food safety statistics?
Materials and Methods.
With regards to the “Study Area”, it is possible to provide some more data? Also considering that Dzorwulu and Jamestown have been selected considering “the socio-economic statuses of their resident” (line 94). Moreover, I cannot understand the link between Accra, which has been selected considering that it is “one of Africa’s emerging mega-cities” (line 92-93) and the selection of two very small communities. What is the link between the selection of one mega-city and two little communities? Please, provide some more details.
In the section “Data Collection”, I suggest the authors providing some more information related to previous studies, which have applied survey in the field of food waste behavior. Several have been conducted, as highlighted by the subsequent review:
Amicarelli, V. and Bux, C. (2020). Food waste measurement toward a fair, healthy and environmental-friendly food system: a critical review. British Food Journal, 123(8), 2021, 2907-2935. 10.1108/BFJ-07-2020-0658
For instance, provide information related to advantages and disadvantages of using questionnaire in the investigation of consumers food waste behavior. Further, I cannot see any information related to the questionnaire: (a) how many questions? (b) on which basis has been developed the questionnaire? (c) are there any research hypotheses? (d) which variables have been investigated? (e) considering the stratified random sampling procedure, which are its biases and its advantages? Etc. Such information are essential to evaluate the research results.
It is a critical issue that no information is provided with reference to the questionnaire composition.
Results: In Table 1, is the “Socio-economic status of household” different from the “monthly income”? How is it possible to evaluate in objective terms “lower-class”, “lower-middle-class”, “middle-class”, etc.? Could you please provide some monetary values, for instance compared to USD? Also, considering that GHÈ» is a very uncommon currency, could you please provide an equivalent in USD?
All relationships investigated in the section “Results” should have been presented in the section “Materials and Methods”. For instance, the authors must declare in the section “Materials and Methods” that the research investigated the relationship between frequency of household food shopping and storage of cassava, plantain, maize, rice, tomato and pepper. It needs a proper justification in the section “Materials and Methods”, not in the section “Results”.
Same applies for the investigation of the relationship between frequency of cooking and storage of cassava, plantain, maize, rice, tomato and pepper. Such investigated variables must be declared in the section “Materials and Methods”. You can provide a graphical tool to highlight the different investigated variables (as illustrated in Table 7).
Same applies for the variable “perception of food storage”.
Discussion / Conclusions: “Discussion” and “Conclusions” are suitable and interesting. The authors match the aims and scope of the research with the related discussion on the topic and provide sufficient answers to the outlined research questions.
Author Response
RESPONSE TO SUGGESTIONS FROM REVIEWER 1
Thank you for reviewing our manuscript. Below are the responses to your reviews or suggestions.
Title:
The manuscript title has been changed to as follows;
“Determinants of Household-Level Food Storage Practices and Outcomes on Food Safety and Security in Accra, Ghana”
Abstract:
No action was taken.
Introduction:
Some more references have been added to lines 34-42.
Some information related to food safety has been added to the manuscript.
Some recent references related to consumer behavior and food storage within households have been added. Nevertheless, some references such as those listed below are key to this study, and therefore have been maintained.
- K. Bond, D. Thilmany, and C. A. Bond, “Direct Marketing of Fresh Produce: Understanding Consumer Purchasing Decisions,” Choices, vol. 21, no. 4, pp. 229–235, 2006,
- Ovuga, J. Boardman, and D. Wassermann, “Prevalence of suicide ideation in two districts of Uganda,” Arch. Suicide Res., vol. 9, no. 4, pp. 321–332, 2005, doi: 10.1080/13811110500182018
- G. Cochran, Sampling Technique. New York: Wiley, 1963.
- Sudakar, “A note on circular systematic sampling,” Sankhya C, pp. 72–73, 1978
- Vyas and L. Kumaranayake, “Constructing socio-economic status indices: How to use principal components analysis,” Health Policy Plan., vol. 21, no. 6, pp. 459–468, 2006, doi: 10.1093/heapol/czl029
- O. Rutstein and K. Johnson, The DHS wealth index: DHS Comparative Reports, 6. Calverton, Maryland: ORC Macro, 2004
A paragraph devoted to the reasons why the authors investigate food security and food safety in Ghana have been included in the manuscript. Some data related to food security and food safety in Ghana have also been provided.
Materials and methods:
Some data have been added to the study area. The scope of the study did not include all the communities within Accra, but the two selected communities in the study (Dzorwulu and Jamestown).
In the section “Data Collection”, information related to previous studies, which have applied survey in the field of household or consumer food safety, waste, security and expenditure have been included.
More information concerning the questionnaire, its composition, number of questions asked, variables investigated and so on have been provided in the manuscript.
Information regarding stratified random sampling and systematic sampling methods, their advantages and biases have been provided.
Results:
Socio-economic status of households is different from their monthly income. Socio-economic status is a measure of the wealth status of households and it can be estimated by using the assets value of a household with the use of principal component analysis (PCA) as described by;
- W. Kabudula, B. Houle, M. A. Collinson, K. Kahn, S. Tollman, and S. Clark, “Assessing Changes in Household Socioeconomic Status in Rural South Africa, 2001–2013: A Distributional Analysis Using Household Asset Indicators,” Soc. Indic. Res., vol. 133, no. 3, pp. 1047–1073, 2017, doi: 10.1007/s11205-016-1397-z.
- Vyas and L. Kumaranayake, “Constructing socio-economic status indices: How to use principal components analysis,” Health Policy Plan., vol. 21, no. 6, pp. 459–468, 2006, doi: 10.1093/heapol/czl029.
- O. Rutstein and K. Johnson, The DHS wealth index: DHS Comparative Reports, 6. Calverton, Maryland: ORC Macro, 2004. Available: https://dhsprogram.com/pubs/pdf/cr6/cr6.pdf
Groupings of the socio-economic status of households have been properly defined and changed in the manuscript to: “Higher-income, Upper-middle-income, Middle-income, Lower-middle-income and Lower-income”.
Considering the study location, the authors will like to maintain the currency unit as Ghana Cedi (GHÈ»).
All relationships investigated in the section “Results” have been presented and justified in the section “Materials and Methods”.
Discussion / Conclusions:
No action was taken.

Reviewer 2 Report
This study aimed to assess the determinants of household-level food storage, ascertain consumers' behavior and perception towards food storage, and assess the effect of household-level food storage on food safety, wastage, food expenditure, and security.
this is a fascinating study and covers one of the big problems that is faced worldwide nowadays. Also, this study is very well written and compiled. The content flow is smooth and easy to read. The experiments were executed in a well-designed manner with experimental controls. The aim and objectives of the study are well addressed. but the conclusion is very long and needs to be shorter.
Author Response
RESPONSE TO SUGGESTIONS FROM REVIEWER 2
Thank you for reviewing our manuscript. Below is the response to your reviews or suggestions.
Considering the scope of the study, the authors will like to maintain the length of the “Conclusion” section as it is.
